# Geometric Phase Effects in Ultracold Chemical Reactions

**Brian K. Kendrick** [1] and **N. Balakrishnan** [2,*]

[1]  Theoretical Division (T-1, MS B221), Los Alamos National Laboratory, Los Alamos, NM 87545, USA
[2]  Department of Chemistry and Biochemistry, University of Nevada, Las Vegas, NV 89154, USA
[*]  Correspondence: naduvala@unlv.nevada.edu; Tel.: +1-702-895-2907

**Abstract:** The role of the geometric phase effect in chemical reaction dynamics has long been a topic of active experimental and theoretical investigations. The topic has received renewed interest in recent years in cold and ultracold chemistry where it was shown to play a decisive role in state-to-state chemical dynamics. We provide a brief review of these developments focusing on recent studies of O + OH and hydrogen exchange in the H + $H_2$ and D + HD reactions at cold and ultracold temperatures. Non-adiabatic effects in ultracold chemical dynamics arising from the conical intersection between two electronic potential energy surfaces are also briefly discussed. By taking the hydrogen exchange reaction as an illustrative example it is shown that the inclusion of the geometric phase effect captures the essential features of non-adiabatic dynamics at collision energies below the conical intersection.

**Keywords:** ultracold chemistry; ultracold molecules; ultracold reactions; geometric phase effect; controlled chemistry

## 1. Introduction

Our description of chemical reactions largely relies on the Born–Oppenheimer (BO) approximation in which the fast-moving electronic degrees of freedom are separately treated from the slow moving nuclear coordinates which largely remain unchanged during an electronic transition. In this adiabatic approximation, the electronic problem is solved for different fixed nuclear configurations and the nuclei are assumed to evolve in a potential field created by the electrons. This separation is the basis of the vast majority of electronic structure and quantum dynamics studies and works amazingly well for many chemical reactions involving electronically ground state atoms and molecules. However, in many cases, the effect of the excited electronic state cannot be neglected, in particular, when there is a conical intersection (CI) between two electronic states, e.g., the ground state and an excited electronic state. In this case, the sign of the electronic wave function changes when the nuclei undergo a closed loop around the CI, or in other words, when the nuclear motion encircles the CI. This requires a corresponding sign change for the nuclear motion wave function to keep the sign of the overall wavefunction unchanged. The sign change is referred to as the geometric phase (GP) and it needs to be taken into account even when the energy of nuclear motion is well below that of the location of the CI. The effect, first pointed out by Longuet-Higgins [1] and Herzberg and Longuet-Higgins [2], has been shown to be important for describing spectra of triatomic molecules [3–7]. However, until recently, its effect on chemical reaction dynamics and state-to-state chemistry has been less obvious despite extensive experimental and theoretical investigations.

For chemical reactions occurring on a single BO electronic potential energy surface (PES) Mead and Truhlar [8] and Mead [9–11] showed that the GP effect can be accounted for by solving a generalized Schrödinger equation for the nuclear motion with an additional vector potential analogous to that of a magnetic solenoid centered at the CI [8]. Berry [12] later showed that the appearance of a vector potential and its associated geometric phase (also known as the Berry phase) is a general consequence

of the adiabatic transport of a quantum state [13]. A number of theoretical studies have been reported in the literature accounting for the GP effect in both bound state [3–7] and scattering calculations of triatomic systems [14–26]. While bound state studies of alkali metal trimers such as $Li_3$ [27], $Na_3$ [28] and transition metal systems like $Cu_3$ [29] showed much better agreement with experimental results when the GP effect is included, experimental verification of the GP effect in a bimolecular chemical reaction has remained elusive [30–35], until recently [36]. The GP effect has also been recently discussed in photochemical processes where GP corrected adiabatic dynamics was shown to yield results in close agreement with an explicit diabatic treatment [37–41].

Almost all of the experimental studies of GP effects in bimolecular chemical reactions have so far been limited to H or D atom exchange reactions in H + HD/D + HD systems at energies close to the CI [30–35]. At these high collision energies, many partial waves contribute and any GP effect present in a partial wave resolved cross section washes out when a summation over all partial waves is carried out to evaluate the total differential or integral cross sections [14,18–26]. However, in a recent experiment on the H + HD→D + $H_2$ reaction [36], it was shown that the GP effect survives the partial-wave summation in the forward scattering differential cross section (DCS) providing the most direct evidence of the GP effect in a chemical reaction. In this work, Yuan et al. [36] reported fast oscillations in the forward scattering state-resolved DCSs for H + HD($v = 0, j = 0$) →D + $H_2(v' = 0, j' = 7)$ and combined D + $H_2(v' = 1, j' = 9)$ and D + $H_2(v' = 2, j' = 3)$ final states. Only, a theoretical treatment that includes the GP effect or a full-nonadiabatic treatment within a diabatic framework that implicitly includes the GP effect was able to account for the fine oscillations in the experimental DCSs [36]. Note that the GP effect is required for a correct treatment within the adiabatic BO separation of nuclear and electronic degrees of freedom. Yuan et al. also demonstrated that the GP effect occurs as a result of the interference between scattering amplitudes along two paths—a direct path that involves one transition state and an indirect path that involves two transition states. The scattering amplitude (and integral cross section) along the indirect path is negligible for collision energies below the energy of the CI (2.75 eV total energy) but becomes comparable for forward scattering at a collision energy of 2.77 eV relative to the $v = 0, j = 0$ rotational level of HD (total energy of 2.99 eV).

In contrast to the thermal energy regime where partial wave summation largely masks the GP effect, the ultracold regime provides a fascinating and fertile ground to investigate the GP effect. In this regime, scattering is dominated by *s*- or *p*-waves depending on the lowest allowed partial wave (*s*-waves for bosons and *p*-waves for fermions) and the effect gets magnified through interference and resonance effects that are prominent in the single partial wave regime. In the ultracold limit the pure *s*-wave regime provides maximum constructive/destructive interference due to isotropic scattering and an effective quantization of the scattering phase shift as predicted by the Levinson's theorem [42]. In a series of papers, Ref. [43–49] we have recently shown that a significant GP effect occurs in both the differential and integral cross sections for state-to-state transitions in O + OH($v = 0 - 1, j = 0$)→H + $O_2(v', j')$, H + HD($v = 4, j = 0$) →HD($v', j'$) + H, D + HD($v = 4, j = 0$) →HD($v', j'$) + D, and H + $H_2(v = 4, j = 0$) →H + p − $H_2(v', j')$ collisions at cold and ultracold temperatures. Although the energy of the $v = 4$ vibrational level of HD (1.905 eV) is well below the energy of the CI, a favorable encirclement of the CI is possible as the reaction becomes effectively barrierless for vibrational levels $v \geq 3$. The scattering amplitudes along the "direct" and "indirect" paths reveal comparable magnitudes leading to strong interference between them. In the following we briefly discuss the mechanism of the GP effect in ultracold chemical reactions taking the O + OH and the hydrogen exchange reactions as illustrative examples.

## 2. Mechanism of GP Effect in Ultracold Chemical Reactions

Althorpe and coworkers [24] have previously shown that the GP and NGP (no geometric phase) scattering amplitudes can be rigorously expressed as a linear combination of the "direct" and "looping" (or indirect) contributions (see illustration in Figure 1 for the O + OH→H + $O_2$ reaction):

$$f_{NGP/GP} = \frac{1}{\sqrt{2}}(f_{direct} \pm f_{loop}), \tag{1}$$

where the minus sign for the GP arises from the sign change due to the GP effect. The differential cross sections are related to the square of the scattering amplitudes. The square modulus of the scattering amplitudes for the NGP and GP calculations becomes

$$|f_{NGP/GP}|^2 = \frac{1}{2}\left(|f_{direct}|^2 + |f_{loop}|^2 \pm 2|f_{direct}||f_{loop}|\cos\Delta\right), \tag{2}$$

where the complex scattering amplitudes $f_{direct} = |f_{direct}|e^{i\delta_{direct}}$ and $f_{loop} = |f_{loop}|e^{i\delta_{loop}}$ and $\Delta = \delta_{loop} - \delta_{direct}$ is the phase difference between the two pathways. For H + HD→H + HD and D + HD→D + HD (or in general, A + AB → A + AB systems) the direct path refers to purely inelastic processes with no atom exchange and the looping path refers to the atom-exchange channel ( such as A + A′B→ A′ + AB and the AB molecule may be in a different ro-vibrational level than A′B). For comparable values of the two scattering amplitudes, i.e., $|f_{loop}| \sim |f_{direct}| = |f|$, Equation (2) becomes $|f_{NGP/GP}|^2 \sim |f|^2(1 \pm \cos\Delta)$. The criterion for maximum constructive and destructive interference can be inferred from the sign and magnitude of $\cos\Delta$. If $\cos\Delta = +1$ then maximum (constructive) interference occurs for the NGP case yielding $|f_{NGP}|^2 \sim 2|f|^2$ and $|f_{GP}|^2 \sim 0$. On the other hand, if $\cos\Delta = -1$ then maximum (constructive) interference occurs for the GP case leading to $|f_{GP}|^2 \sim 2|f|^2$ and $|f_{NGP}|^2 \sim 0$. This maximum constructive/destructive interference can occur in the ultracold regime because the phase shifts $\delta_{direct}$ and $\delta_{loop}$ are effectively quantized, i.e., $\delta_{direct} = n_{direct}\pi$ and $\delta_{loop} = n_{loop}\pi$ (Levinson's theorem [42]) where $n_{direct}$ and $n_{loop}$ are the number of bound states supported by the potential well along the direct and looping paths. This yields, $\Delta = n\pi$ where $n = n_{loop} - n_{direct}$. Thus, the reaction can be turned on or off depending simply on the sign of the interference term (since $|\cos\Delta| \sim 1$). If one of the scattering amplitudes is much greater than the other, $|f_{loop}|^2 >> |f_{direct}|^2$ or $|f_{direct}|^2 >> |f_{loop}|^2$, then Equation (2) becomes $|f_{NGP/GP}|^2 \sim |f_{loop}|^2/2$ or $|f_{NGP/GP}|^2 \sim |f_{direct}|^2/2$. The GP effect vanishes in this case and the interference term containing $|\cos\Delta|$ plays no role. In the high partial wave limit (high collision energies), the interference term averages out to zero ($\cos\Delta \sim 0$) and there is no GP effect. The interference mechanism of the GP effect discussed here (where the Levinson's theorem is applied for the behavior of the scattering phase shift) is specific to the ultracold *s*-wave limit.

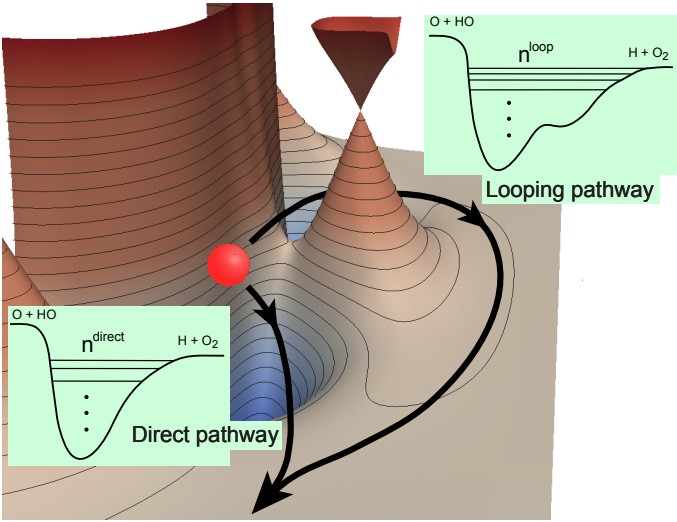

**Figure 1.** Direct and looping pathways for the O + OH reaction illustrating the number of bound states supported in the effective potentials along each pathway [43]. Reproduced with permission from Kendrick et al. [43].

### 3. Quantum Scattering Method

Reactive scattering calculations are performed in hyperspherical coordinates based on the formalism developed by Pack and Parker [14,50,51]. These coordinates include three internal coordinates $(\rho, \theta, \phi)$ and three external coordinates (three Euler angles $\alpha, \beta, \gamma$). In the approach based on the adiabatically-adjusting principle axis hyperspherical (APH) coordinates of Pack and Parker, the hyperradius $(\rho)$ is divided into two regions: an inner region where rearrangement scattering occurs and an outer region where the coupling between different arrangement channels vanishes. The APH coordinates are used in the inner region which treat all three atom-diatom arrangement channels (A + BC, AB + C, AC + B channels in a triatomic molecule ABC) equivalently while the Delves hyperspherical coordinates are used in the outer region. The APH and Delves coordinates differ in the choice of the hyperangles $(\theta, \phi)$ while the hyperradius remains the same. The Delves coordinates allow convenient mapping to Jacobi coordinates in different arrangement channels in the asymptotic region. The hyperradius in both regions is divided into a large number of sectors (typically about 100 sectors in each region) and at each sector the overall nuclear motion wave function is expanded in terms of hyperspherical surface functions [50,51]. In the inner region, the surface functions are expanded a hybrid basis set consisting of a discrete variable representation (DVR) for the hyperangle $\theta$ and a finite basis representation (FBR) for the azimuthal angle $\phi$ [51]. In the outer region, the overall wave function at each sector is expanded in terms of asymptotic ro-vibrational wave functions of the diatomic fragments. The surface function Hamiltonian in the inner region is diagonalized using a parallel implementation of the implicitly restarted Lanczos algorithm [51]. The hybrid FBR/DVR method is computationally efficient and it permits sequential diagonalization truncation to dramatically reduce the size of the Hamiltonian matrix. This is particularly important in keeping the matrix size small for large values of the total angular momentum (vector sum of the orbital angular momentum of the collision complex and the rotational angular momentum of the dimer) quantum number $J$.

The GP effect is included only in the inner region (relevant for triatomic configurations) through the vector potential approach [19,20]. The log-derivative method of Johnson is employed to propagate the radial equation in $\rho$ subjected to appropriate boundary conditions. At the last sector of the hyperradius in the inner region, the log-derivative matrix is transformed to Delves coordinates to continue the propagation to the last sector $(\rho = \rho_{max})$ where asymptotic boundary conditions are applied for the different channels to extract the scattering $S$-matrix. Relevant state-to-state cross sections are evaluated from the $S$-matrix using standard formulas.

In the ultracold regime scattering is dominated by $s$-waves and in this limit Wigner threshold laws apply for elastic, inelastic and reactive cross sections. Inelastic and reactive cross sections diverge inversely as the collision velocity leading to a finite value for the limiting zero-temperature rate coefficients. The limiting rate coefficients can be expressed as $k_{vj} = 4\pi\hbar\beta_{vj}/\mu$ where $\beta_{vj}$ is the imaginary part of the complex scattering length, $a_{vj} = \alpha_{vj} - i\beta_{vj}$ [52] and $\mu$ is the reduced mass of the system. This limiting value is equivalent to the non-thermal rate coefficient obtained by multiplying the cross sections in the Wigner limit with the relative velocity. The rate coefficients presented here correspond to the non-thermal values.

### 4. Results

#### 4.1. O + OH→H + O$_2$ Reaction

The O + OH→H + O$_2$ reaction is one of the most important reactions in atmospheric and combustion chemistry and is believed to play a key role in oxygen chemistry in the interstellar medium. It is widely used as a benchmark for complex-forming reactions in chemical dynamics and has been the topic of numerous experimental and theoretical investigations. Previous bound state calculations have shown significant differences in bound state energies of the HO$_2$ radical when the geometric phase effect is included [19,20]. Our recent study showed significant GP effects in state-to-state rotationally

resolved rates for this reaction occurring from the $v = 0, j = 0$ and $v = 1, j = 0$ initial states of the OH molecule [43,44]. In Figure 2 we show state-to-state cross sections for the O + OH($v = 0, j = 0$) → H + O$_2$($v', j'$) reaction for $v' = 0, j' = 1$; $v' = 0, j' = 3$; $v' = 1, j' = 1$; $v' = 1, j' = 15$; $v' = 3, j' = 1$; and $v' = 3, j' = 3$, respectively, in panels (a)–(f). It is seen that the rate coefficients for the GP and NGP cases differ by a factor of three to an order of magnitude for different final states in the ultracold regime but they become nearly identical for collision energies above 1 K. As the contributions from higher partial waves become important for collision energies above 0.01 K the GP effect gets washed out due to partial wave summation. The GP effect also becomes less pronounced in the vibrationally resolved (summed over $j'$ levels) and total (summed over $v', j'$ levels) reaction rates. This is illustrated in Figure 3 where the *J*-resolved and *J*-summed total reactive rates are shown as a function of the collision energy. Thus, based on these results, a measurement of state-to-state rate coefficients in the single partial wave regime could provide the most direct evidence of GP effect in this reaction.

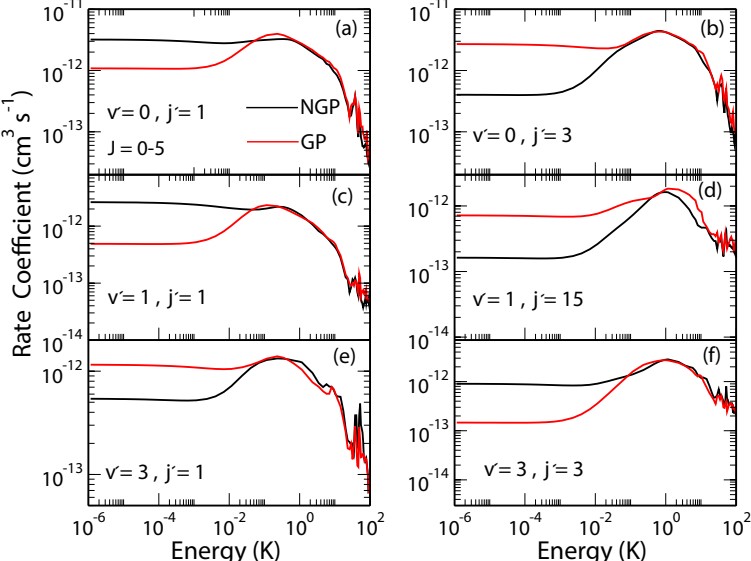

**Figure 2.** State-to-state rate coefficients for the O + OH($v = 0, j = 0$) → H + O$_2$($v', j'$) reaction for different $v', j'$ states of the O$_2$ molecule plotted as functions of the collision energy. The $v', j'$ states are specified in each panel. Reproduced with permission from Hazra et al. [44].

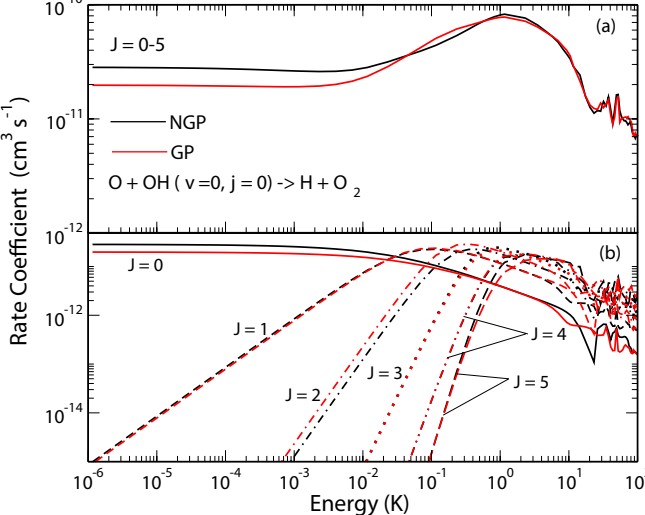

**Figure 3.** Rate coefficients for the O + OH($v = 0, j = 0$) → H + O$_2$ reaction (summed over all $v', j'$ states) plotted as functions of the collision energy. Panel (**a**) summed over the total angular momentum quantum number $J = 0 - 5$, and (**b**) contribution from different values of $J$. Reproduced with permission from Hazra et al. [44].

### 4.2. Hydrogen Exchange Reaction

We have carried out extensive analysis of GP effects in H + H$_2$ , H + HD and D + HD reactions [45–49]. Atom exchange in these systems involves an energy barrier but the barrier decreases with vibrational excitation of the molecule and the reaction becomes barrierless for vibrational levels $v > 3$ for both molecules. Indeed, vibrationally adiabatic potentials for $v = 4$ and higher vibrational levels of H$_2$ and HD depict a potential well [48]. These potential wells along the direct and exchange paths may feature different bound state structure and scattering phase shifts. In H + HD and D + HD collisions, H or D atom exchange can lead to chemically distinct products (D + H$_2$ and H + D$_2$) or atom exchange such as H + H'D→H' + HD. Thus rovibrational level changes of HD can occur through a purely inelastic channel (no-atom exchange) or a reactive pathway (through atom exchange). The GP and NGP wave functions can be expressed as a linear combination of this direct and exchange channels similar to the direct and looping contributions given in Equation (2). The computations can be separately carried out for even and odd exchange symmetry contributions (exchange of the identical H or D atoms). Figure 4 shows a comparison of cross sections for the GP and NGP cases for both even and odd exchange symmetries for the D + HD($v = 4, j = 0$) → D + HD($v' = 3, j' = 0$) reaction as a function of the scattering angle and the collision energy. It is seen that constructive/destructive interference between the direct and exchange pathways leads to complete enhancement (suppression) of the NGP (GP) cross sections for the even exchange symmetry while just the opposite occurs for the odd exchange symmetry. The overall cross section is the sum of the even and odd exchange symmetry contributions weighted by appropriate nuclear spin statistical factors (2/3 for even and 1/3 for odd).

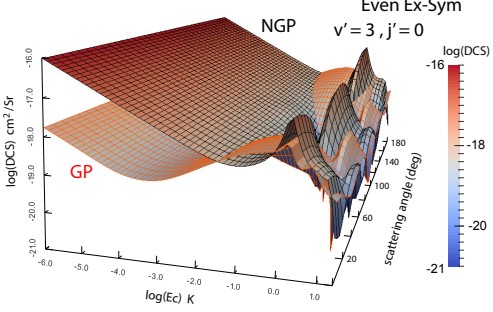

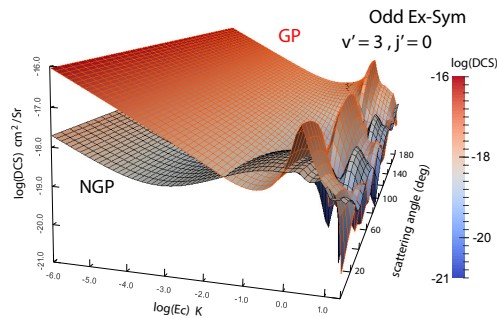

**Figure 4.** Differential cross sections for the D + HD($v = 4, j = 0$) → D + HD($v' = 3, j' = 0$) reaction plotted as functions of the collision energy and scattering angle. Left panel: even exchange symmetry; right panel: odd exchange symmetry. Reproduced with permission from Kendrick et al. [46].

The effect is more dramatic for the H + H$_2$ reaction [47]. Figure 5 shows total and *J*-resolved rate coefficients for the H + H$_2(v = 4, j = 0) \rightarrow$H + p − H$_2$ reaction. It is seen that the GP and NGP rates summed over all final ro-vibrational levels of para-H$_2$ differ by an order of magnitude with the GP rates being enhanced by constructive interference in the ultracold regime. Another striking aspect of the results presented in Figure 5 is the resonant enhancement of the NGP rate near 1–3 K which arises primarily from *J* = 1 (partial wave *l* = 1). Thus, the absence of the *l* = 1 shape resonance in the GP results and the enhancement of the GP rates in the ultracold regime provide signatures of GP effect that may be detectable in an experiment.

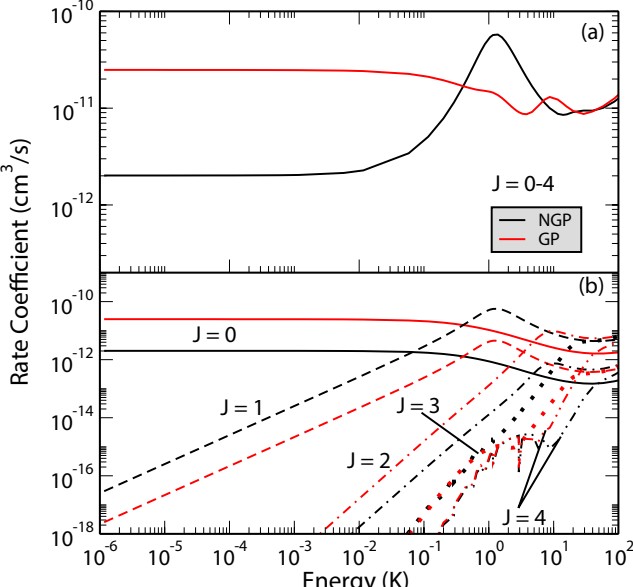

**Figure 5.** Rate coefficients for the H + H$_2(v = 4, j = 0) \rightarrow$ H + p − H$_2$ reaction plotted as functions of the collision energy. Panel (**a**) summed over the total angular momentum quantum number *J* = 0 − 4, and (**b**) contribution from different values of *J*. Reproduced with permission from Kendrick et al. [47].

## 5. Non-Adiabatic Effects

For collision energies at or above the CI the effect of the excited electronic state (or states) must be explicitly taken into account. This requires a treatment that goes beyond the BO approximation and needs to include non-adiabatic interactions between the lower and upper electronic states. Kendrick has recently developed the general formalism for non-adiabatic dynamics in hyperspherical coordinates and applied it to the hydrogen exchange reactions [53]. For a system of two coupled electronic states with electronic wave functions $\phi_1$ and $\phi_2$, the Schrödinger equation may be written as

$$\left[ \frac{\hbar^2}{2\mu} \begin{pmatrix} -i\nabla_{\mathbf{R}} & -\mathbf{A}_{12} \\ -\mathbf{A}_{21} & -i\nabla_{\mathbf{R}} \end{pmatrix} \begin{pmatrix} -i\nabla_{\mathbf{R}} & -\mathbf{A}_{12} \\ -\mathbf{A}_{21} & -i\nabla_{\mathbf{R}} \end{pmatrix} + \begin{pmatrix} V_1 & 0 \\ 0 & V_2 \end{pmatrix} \right] \begin{pmatrix} \psi_1(\mathbf{R}) \\ \psi_2(\mathbf{R}) \end{pmatrix} = E \begin{pmatrix} \psi_1(\mathbf{R}) \\ \psi_2(\mathbf{R}) \end{pmatrix}, \quad (3)$$

where $V_1$ and $V_2$ are the adiabatic PESs for the lower and upper electronic states and $\mathbf{A}_{12} = \mathbf{A}_{21} = i\langle \phi_1 | \nabla_{\mathbf{R}} | \phi_2 \rangle$ are the derivative couplings (non-adiabatic couplings). The gradient operator $\nabla_{\mathbf{R}}$ denotes the derivative with respect to all nuclear coordinates **R**. Alternatively, one may define a diabatic representation (strictly, a quasi-diabatic representation since we have restricted to just two electronic states), with the adiabatic to diabatic transformation given by

$$\begin{pmatrix} \tilde{\psi}_1(\mathbf{R}) \\ \tilde{\psi}_2(\mathbf{R}) \end{pmatrix} = \begin{bmatrix} \cos\alpha(\mathbf{R}) & -\sin\alpha(\mathbf{R}) \\ \sin\alpha(\mathbf{R}) & \cos\alpha(\mathbf{R}) \end{bmatrix} \begin{pmatrix} \psi_1(\mathbf{R}) \\ \psi_2(\mathbf{R}) \end{pmatrix}. \quad (4)$$

where $\tilde{\psi}_1(\mathbf{R})$ and $\tilde{\psi}_2(\mathbf{R})$ are the nuclear wave functions in the diabatic representation and $\alpha(\mathbf{R})$ is the mixing angle between the two adiabatic electronic states. In the diabatic representation, the derivative couplings are zero but the potential energy matrix acquires off-diagonal elements. The Schrödinger equation in the diabatic representation becomes

$$-\frac{\hbar^2}{2\mu} \begin{pmatrix} \nabla_{\mathbf{R}}^2 & 0 \\ 0 & \nabla_{\mathbf{R}}^2 \end{pmatrix} \begin{pmatrix} \tilde{\psi}_1(\mathbf{R}) \\ \tilde{\psi}_2(\mathbf{R}) \end{pmatrix} + \begin{pmatrix} V_{11} & V_{12} \\ V_{21} & V_{22} \end{pmatrix} \begin{pmatrix} \tilde{\psi}_1(\mathbf{R}) \\ \tilde{\psi}_2(\mathbf{R}) \end{pmatrix} = E \begin{pmatrix} \tilde{\psi}_1(\mathbf{R}) \\ \tilde{\psi}_2(\mathbf{R}) \end{pmatrix}. \tag{5}$$

The mixing angle $\alpha(\mathbf{R})$ can be chosen as:

$$-\alpha(\mathbf{R}) = \frac{1}{2} \arctan \frac{2V_{12}}{V_{11} - V_{22}}. \tag{6}$$

The matrix that transforms the electronic wave functions from the adiabatic to diabatic representation also transforms the diabatic potential matrix to the adiabatic representation:

$$V_{1,2} = \frac{1}{2}(V_{11} + V_{22}) \pm \frac{1}{2}\sqrt{(V_{11} - V_{22})^2 + 4V_{12}^2}. \tag{7}$$

The adiabatic and diabatic representations are formally equivalent, however, the derivative couplings ($\mathbf{A}_{nm}$) which appear in the adiabatic Hamiltonian are singular at the CI. These singularities are very difficult to treat numerically, so the smooth non-singular diabatic Hamiltonian is used instead. The diabatic approach also implicitly accounts for the GP effect.

Figure 6 plots the total rate coefficient as a function of collision energy for the D + HD($v = 4$, $j = 0$) → D + HD reaction computed using the two-state diabatic representation (Equation (5)) [53]. For comparison, our previous GP results which were computed using the adiabatic representation with a vector potential on a single ground electronic state BO PES are also plotted in red [45,48]. The results for even (odd) exchange symmetry are plotted using solid (dashed) curves, respectively. The total energy of ultracold D + HD($v = 4$, $j = 0$) collisions is approximately 1.9 eV which is well below the energy of the CI ($\approx$2.7 eV). Thus, the non-adiabatic couplings to the excited electronic state are expected to be small and the results from the two-state diabatic calculation (which implicitly includes the GP) are expected to agree with the adiabatic + GP results. From Figure 6 we see that this is indeed the case, the black and red curves are essentially identical over the entire energy range ($E_c = 1\,\mu\text{K} - 100\,\text{K}$). The $l = 1$ and $l = 2$ shape resonances (bumps) near 2 K and 7 K, respectively, are also well reproduced by both sets of calculations. However, for total energies which lie above the energy of the CI, we expect that the couplings with the excited electronic state will become important and the two-state diabatic results will be different than the adiabatic + GP results. The high energy regime was recently investigated for the H + H$_2$ reaction with high vibrational excitation of H$_2$($v = 4 - 8$) [54]. For the ultracold H + H$_2$($v = 8$, $j = 0$) collisions, the total energy is approximately 3.6 eV which is significantly above the energy of the CI. Figure 7 plots the total rate coefficient as a function of collision energy for the H + H$_2$($v = 8$, $j = 0$) → H + p − H$_2$ reaction computed using the two-state diabatic representation (in black), the adiabatic with GP (in red), and the adiabatic without GP (in blue). The diabatic and adiabatic + GP are in good agreement at ultracold collision energies and for energies above 0.2 K. However, for intermediate energies near the $l = 1$ shape resonance at 0.01 K, we see significant differences between the diabatic and adiabatic + GP results. As expected, the adiabatic (NGP) rate coefficient is significantly different from both the diabatic and adiabatic + GP results: it is suppressed by a factor of three at ultracold collision energies, and the shape resonance near 0.01 K is enhanced while the shape resonance near 0.5 K is suppressed. Figure 7 confirms that a coupled two-state diabatic calculation is required for total energies which lie significantly above the energy of the CI and that both GP and other non-adiabatic effects are important.

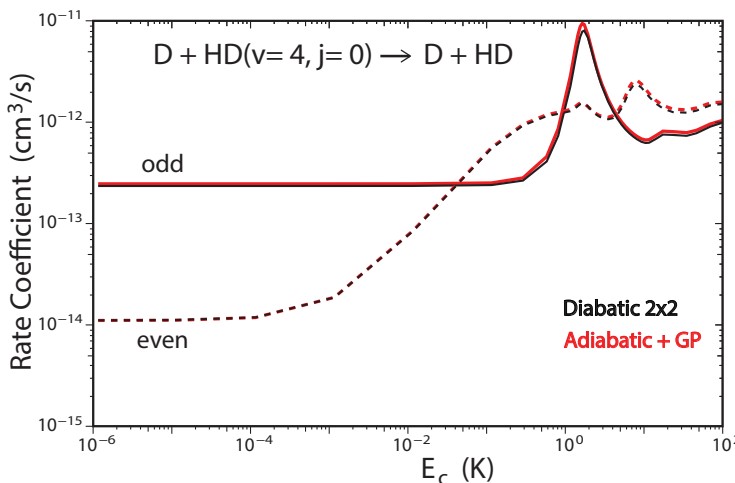

**Figure 6.** Rate coefficients for the $D + HD(v = 4, j = 0) \rightarrow D + HD$ reaction (summed over all final $v'$, $j'$ states) plotted as functions of collision energy. Reproduced with permission from Kendrick [53].

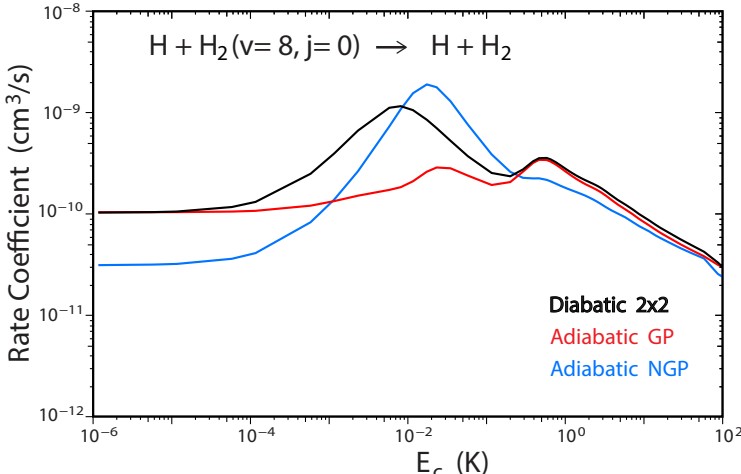

**Figure 7.** Rate coefficients for the $H + H_2(v = 8, j = 0) \rightarrow H + p - H_2$ reaction (summed over all final $v'$, $j'$ states) plotted as functions of the collision energy. Reproduced with permission from Kendrick [54].

A recent study of the ultracold $H + H_2$ reaction also revealed a new experimentally detectable signature of the GP effect which manifests itself in the behavior of the two prominent shape resonances as a function of increasing $H_2$ vibrational excitation [54]. Figure 8 plots the total rate coefficient for $H + H_2(v = 4 - 8) \rightarrow H + p - H_2$ as a function of the collision energy computed using the two-state diabatic representation. For initial vibrational quanta $v = 4$ and 5, we see that the two shape resonances near 1 K and 8 K are relatively suppressed (small bumps). However, for $v = 6$ the two shape resonances become significantly pronounced. As the initial vibrational excitation of $H_2$ is increased further to $v = 7$ and 8, the shape resonances continue to become more pronounced: their lifetimes increase and their energies shift significantly lower. For example, the lifetime for the $l = 1$ shape resonance for $v = 8$ is two orders of magnitude larger than that for $v = 4$ [54]. This behavior is consistent with a two-body universal collision model [55]. The resonances associated with large $v$ lie deeper behind the centrigual barrier which results in a lower resonance energy and longer lifetime due to the decreased tunneling rate through the wider effective barrier. In contrast, for small $v$ the resonances happen to lie near the top of the centrifugal barrier which results in a higher resonance energy and much shorter lifetime. Calculations which ignore the GP show an entirely different shape resonance behavior. Namely, the $l = 1$ shape resonance dominates for all values of $v$ (e.g., see the blue curve in Figure 7).

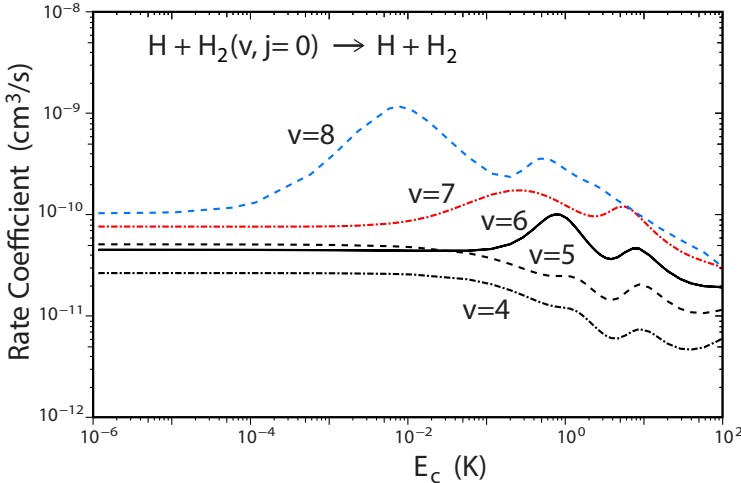

**Figure 8.** Rate coefficients for the $H + H_2(v, j = 0) \to H + p - H_2$ reaction (summed over all final $v'$, $j'$ states) plotted as functions of the collision energy for different initial vibrational quantum numbers $v = 4$ to $8$ computed using the two-state diabatic representation. Reproduced with permission from Kendrick [54].

## 6. Conclusions

We have discussed how the geometric phase effect manifests in ultracold atom-diatom chemical reactions and how it gets magnified or suppressed through constructive and destructive interferences between wave functions that encircle the conical intersection between two electronic states. Illustrative results are presented for the O + OH and hydrogen exchange reactions. Explicit scattering calculations including the excited electronic state show that the geometric phase effect captures essential features of non-adiabatic dynamics at total energies below the conical intersection. However, for total energies above the conical intersection, other non-adiabatic effects also become important in addition to the geometric phase and a coupled two-electronic state calculation is required. Large values of total energy can be accessible even for ultracold atom–diatom collisions through the vibrational excitation of the reactant diatomic molecule. As the reactant vibrational excitation is increased, interesting non-adiabatic and geometric phase effects on the shape resonances are observed in the hydrogen exchange reaction. Specifically, the resonance energies shift lower and the lifetimes increase dramatically which are consistent with quasi-bound states which lie more deeply behind the centrifugal energy barrier associated with non-zero values of total angular momentum.

For ultracold reactions involving alkali metals (i.e., K + KRb, Li + LiNa, Na + NaK, . . . ) the reactions are typically exoergic, exhibit a deep potential well, and contain a low-lying conical intersection which is energetically accessible even for ultracold collisions involving ground state reactants ($v = 0, j = 0$). Thus, for these ultracold reactions a coupled two-electronic state calculation is required in order to account for both non-adiabatic and geometric phase effects. Our on going studies of ultracold chemical reactions will be investigating these systems using the newly developed non-adiabatic quantum reactive scattering methodology in hyperspherical coordinates [53].

**Author Contributions:** conceptualization, B.K.K. and N.B.; methodology, B.K.K. and N.B.; software, B.K.K.; validation, B.K.K.; formal analysis, B.K.K. and N.B.; investigation, B.K.K. and N.B.; resources, B.K.K. and N.B.; data curation, B.K.K.; writing—original draft preparation, B.K.K. and N.B.; writing—review and editing, B.K.K. and N.B.; Visualization, B.K.K.; supervision, B.K.K. and N.B.; project administration, B.K.K. and N.B.; funding acquisition, B.K.K. and N.B.

**Funding:** B.K.K. acknowledges that part of this work was done under the auspices of the US Department of Energy and funded by Project No. 20170221ER of the Laboratory Directed Research and Development Program at Los Alamos National Laboratory. Los Alamos National Laboratory is operated by Triad National Security, LLC, for the National Nuclear Security Administration of the U.S. Department of Energy (Contract No. 89233218CNA000001). N.B. acknowledges support from National Science Foundation grant No. PHY-1806334.

**Conflicts of Interest:** The authors declare no conflict of interest.

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
