# Peer review of "Geometric Phase Effects in Ultracold Chemical Reactions"

_atoms, doi:10.3390/atoms7030065_

Round 1
Reviewer 1 Report
This review article summarizes the research done on the topic of geometric phase during the last 15 years or so, with emphasis on the work of Kendrick, Balakrishnan and co-workers. This is a nice summary, clearly written and aimed at the broad audience. In particular, the emphasis is made on those predictions of theory that are measurable, and thus can be verified experimentally. The recent literature is also reviewed with enough care. If the authors account for the minor suggestions listed above, the paper can be published without further review.
Line 29: Reference [3] is corrupted. The title of one paper is combined with citation information for the other. Please correct this, by including both references. Both of them seem to be relevant.
Lines 37-39: Please note that Reference [13] is not cited in the manuscript at all, but is included in the list of references. It should probably be cited in the introductory section.
Line 54: The same formula is typed twice, for whatever reason, separated by the &. Is it a misprint? Or a different equation thought to be included? Please correct this issue.
Line 71: Note that reference [47] appear before the references [40-46] are given. Please swap them.
Lines 83-87: This sentence sound misleading. The part in parenthesis “purely inelastic excluding elastic scattering” makes it sound almost opposite the desirable meaning. I would recommend rephrasing it. May be, splitting onto two separate sentences.
Line 86: Use prime symbol for one of the A, instead of red color.
Lines 87, 88: The sign “~” (approximately equal) is more appropriate here, rather than exactly equal “=”.
Lines 90-100: In this discussion as a whole, what would happen if exactly the same number of states is supported by the wells in both direct and looping passes? For example, what if the PES is shallow and the vibrational states are extremely delocalized, such that exactly the same states are cross over along both passes? What is predicted for this case, I wonder??
Lines 187-188: This sentence almost sounds line predictions of the NGP method could be verified experimentally. I would recommend using the word “artificial” or “erroneous” in the description of some of these results, to make clear which of these can be observed in the experiment, and which should not be there.
Line 203: The future tense “we will use” should probably be replaced, since the work done in the past is reviewed.
Figure 6: It has too much text included in the picture. I would recommend removing the words “solid” and “dashed”, and shifting the words “odd sym” and “even sym” closer to the corresponding curves, e.g., in the left side of the figure.
Lines 227-228: Same story: use “artificial” and/or “erroneous”.
References: I would encourage authors to include few more recent relevant references (please note that I am not associated with this group, but their research seems to be very relevant to the topics addressed in this review): Li, J.; Joubert-Doriol, L.; Izmaylov, A. F. Geometric Phase Effects in Excited State Dynamics Through a Conical Intersection in Large Molecules: N-Dimensional Linear Vibronic Coupling Model Study. J. Chem. Phys. 2017, 147 (6), 064106; Izmaylov, A. F.; Li, J.; Joubert-Doriol, L. Diabatic Definition of Geometric Phase Effects. J. Chem. Theory Comput. 2016, 12 (11), 5278–5283.
Reviewer 2 Report
The authors review the geometric phase effect in cold and ultra cold chemical reactions focusing on the H+HD, D+HD, H+H2 and O+OH collisions systems, discussing the effect on the inelastic cross sections / rates, atom exchange and in one case the effect of non-adiabatic coupling.
The authors also suggest various collisions processes that could be studied by experiments in order to see the effect of the geometric phase.
This article is very well written, results are presented clearly, and the theoretical details are well explained and cited.
I am very happy to recommend this article for publication.
Listed below are a couple of additional thoughts for the authors to consider:
- In Fig. 2 are the resonances at high energies shape resonances?
These resonances should be discussed considering that the geomtric phase
has a considerable effect on the HO2 radical bound state energies.
Can these high energy resonance positions be used to see the effect of the geometric phase?
Perhaps it is worth suggesting experiments probe this collision system at these energies.
- Although the H+HD collision system is discussed alongside the D+HD results
no results are explicitly presented for H+HD collisions.
Perhaps this warrants rewording of the abstract to exclude the “H+HD” collision system.
There are just a few minor formatting and consistency issues that should be taken care of before publication:
- Throughout the text the acronym CI and GP are not consistently used (see line 127, and Section 6).
- The placement and spacing of the citations, equation numbers and figure numbers need to be reviewed throughout the whole text so that they are consistent.
See for example lines 151, 158, in contrast to line 213 (Fig. numbering),
line 208 (citations placement), and lines 98, 173 in contrast with line 206 (equation numbering).
Figures 2 and 3 the total angular momentum quantum number J is not defined in the text and should be included between lines 158-161 (or before).
- Figures 3 and 5 description of (b) should be replaced from “for different values of J” to “contribution from different values of J”.
